# Transcriptome Profiling of a Common Mistletoe Species Parasitizing Four Typical Host Species in Urban Southwest China

**DOI:** 10.3390/genes13071173

**Published:** 2022-06-29

**Authors:** Jingge Kuang, Yufei Wang, Kangshan Mao, Richard Milne, Mingcheng Wang, Ning Miao

**Affiliations:** 1Key Laboratory of Bio-Resource and Eco-Environment of Ministry of Education, College of Life Sciences, Sichuan University, Chengdu 610064, China; kjg1106@163.com (J.K.); yufei8828@gmail.com (Y.W.); maokangshan@163.com (K.M.); 2Institute of Molecular Plant Sciences, The University of Edinburgh, Edinburgh EH9 3JH, UK; r.milne@ed.ac.uk; 3Institute for Advanced Study, Chengdu University, Chengdu 610064, China; wangmcyao@163.com

**Keywords:** mistletoe, host selection, hemiparasite, *Taxillus nigrans*, RNA-seq, WGCNA

## Abstract

Comparing gene expressions among parasitic plants infecting different host species can have significant implications for understanding host–parasite interactions. *Taxillus nigrans* is a common hemiparasitic species in Southwest China that parasitizes a variety of host species. However, a lack of nucleotide sequence data to date has hindered transcriptome-level research on *T. nigrans*. In this study, the transcriptomes of *T. nigrans* individuals parasitizing four typical host species (*Broussonetia papyrifera* (Bpap), a broad-leaved tree species; *Cryptomeria fortunei* (Cfor), a coniferous tree species; *Cinnamomum septentrionale* (Csep), an evergreen tree species; and *Ginkgo biloba* (Gbil), a deciduous-coniferous tree species) were sequenced, and the expression profiles and metabolic pathways were compared among hosts. A total of greater than 400 million reads were generated in nine cDNA libraries. These were de novo assembled into 293823 transcripts with an N50 value of 1790 bp. A large number of differentially expressed genes (DEGs) were identified when comparing *T. nigrans* individuals on different host species: Bpap vs. Cfor (1253 DEGs), Bpap vs. Csep (864), Bpap vs. Gbil (517), Cfor vs. Csep (259), Cfor vs. Gbil (95), and Csep vs. Gbil (40). Four hundred and fifteen unigenes were common to all six pairwise comparisons; these were primarily associated with Cytochrome P450 and environmental adaptation, as determined in a KEGG enrichment analysis. Unique unigenes were also identified, specific to Bpap vs. Cfor (808 unigenes), Bpap vs. Csep (329 unigenes), Bpap vs. Gbil (87 unigenes), Cfor vs. Csep (108 unigenes), Cfor vs. Gbil (32 unigenes), and Csep vs. Gbil comparisons (23 unigenes); partial unigenes were associated with the metabolism of terpenoids and polyketides regarding plant hormone signal transduction. Weighted gene co-expression network analysis (WGCNA) revealed four modules that were associated with the hosts. These results provide a foundation for further exploration of the detailed molecular mechanisms involved in plant parasitism.

## 1. Introduction

Parasitic plants are a diverse group of 4750 species that obtain water, mineral nutrients, and carbon from other plants using a specialized feeding organ called a haustorium; within the angiosperms, parasitism has evolved independently at least twelve times [1,2]. The majority of parasitic plants are hemiparasites, which feed directly on other plants but also maintain green leaves and photosynthesize [3]. Many host characteristics interact to determine parasite performance, including nitrogen content [4], carbon content [5], the presence of secondary compounds [6,7]), host condition [8], defenses and immunity [9], biomass [10], and genotype [11]. This complexity has impeded research into hemiparasite host range evolution, particularly as many of the host variables are confounded (depending on the host species).

The ecological significance of parasitic plants depends on the host plant’s preferences and specificity; most mistletoe species have a wide host range and may attach to a diversity of host plants belonging to co-occurring plant species. For instance, *Parentucellia viscosa*, *Rhinanthus minor*, and *Triphysaria versicolor* are known to parasitize 27, 50, and 25 different plant species, respectively [12,13]. However, among all possible host species, each mistletoe species may have a preferred host(s), for which it shows higher specificity than other host species [14]. *Santalum acuminatum* performs better (as measured by its biomass, percent cover, and haustoria abundance) when parasitizing nitrogen-fixing woody plants [15]; the haustoria of Thesium chinense are also larger when parasitizing the Fabaceae species as compared to other host species [16]. This enhanced performance on leguminous hosts appears to result from a combination of less effective anti-parasitic defenses and the availability of sufficient amounts of easily absorbed nitrogen-containing compounds within the host, rather than being a direct consequence of nitrogen fixation [17]. In addition, germination stimulants also affect parasite host selectivity [6,18].

The Loranthaceae family comprises approximately 70 genera and 950 species, all of which are hemiparasites [19]. Within the family, *Taxillus nigrans* is a hemiparasitic species that retains the ability to photosynthesize while obtaining water, carbon, and nutrients from host plants via the haustoria [19,20]. *T. nigrans* can only be propagated by the seeds. It produces fleshy berries that are eaten by birds, promoting seed dispersal; seeds rapidly germinate when deposited on a suitable host, forming new haustoria and drilling into the host’s cortex [21]. The semiparasitism seen in *T. nigrans* is similar to that of other Loranthaceae species, such as *Agelanthus natalitius* and *Struthanthus* aff. *polyanthus* [19,22]. *T. nigrans* can parasitize 51 host species [23], causing the death of branches or the entire plant of the host. However, very little is known about the mechanisms driving host species preference.

With the advent of next-generation sequencing (NGS), the de novo assembly and analysis of parasitic plant transcriptomes have offered important insights into gene expression differences related to host species identity [24]. Here, RNA-seq was used to generate transcriptome profiles for *T. nigrans* individuals parasitizing four different host species (*Broussonetia papyrifera*, *Cinnamomum septentrionale*, *Cryptomeria fortune*, and *Ginkgo biloba*) to reveal underlying molecular differences. A fully annotated transcriptome was obtained and used as a reference to examine expression differences associated with each host species. These results will be helpful in understanding gene expression in the mistletoe species colonizing different host species.

## 2. Materials and Methods

### 2.1. Sample Collection

In July 2016, fresh *T. nigrans* were collected from four different host species (Figure 1), namely, *Broussonetia papyrifera* (Bpap), *Cinnamomum septentrionale* (Csep), *Cryptomeria fortunei* (Cfor), and *Ginkgo biloba* (Gbil), on the Wangjiang Campus of Sichuan University, Chengdu, Sichuan, China. *T. nigrans* has 41 known host species [21]; the host species selected for this study included two deciduous broad-leaved species (Gbil and Bpap), a deciduous coniferous species (Cfor), and an evergreen broad-leaved species (Csep). For the leaf samples, three biological replicates were collected per host species. Samples were immediately frozen in liquid nitrogen and then stored in a −80 ${}^{\circ }$C refrigerator prior to RNA extraction. All samples were used for RNA extraction and RNA-Seq library construction.

### 2.2. RNA-Seq Library Preparation and Sequencing

Total RNA was extracted from collected leaves using Guanidine thiocyanate (Sigma, 50983, Beijing, China)-Chloroform (Sigma, 472476, Beijing, China) according to the manufacturer’s instructions. For each host species, two hundred milligrams of leaf tissue was ground in liquid nitrogen to extract total RNA. The total RNA samples were then treated with a DNA-free${}^{\mathrm{TM}}$ DNA Removal Kit (Ambion, AM1906, Shanghai, China) to remove contaminating genomic DNA. RNA purity was checked using a Nano-Photometer spectrophotometer (IMPLEN, Westlake Village, CA, USA). Before cDNA synthesis, RNA concentrations were measured using a Qubit RNA Assay Kit (Life Technologies, Q32852, Shanghai, China), and RNA integrity was assessed using the RNA Nano 6000 Assay Kit for the Agilent Bioanalyzer 2100 system (Agilent Technologies, Santa Clara, CA, USA).

A total of 3 μg of RNA per sample was used as input material for sample preparation. Twelve sequencing libraries (i.e., three replicates per host species) were generated using the NEBNext${}^{{}^{®}}$ Ultra${}^{\mathrm{TM}}$ RNA Library Prep Kit for Illumina (NEB, Beijing, China) following the provided protocol. Clustering of the index-coded samples was then performed with the cBot Cluster Generation System using a TruSeq PE Cluster Kit v3-cBot-HS (Illumina, Shanghai, China) according to the manufacturer’s instructions. After cluster generation, the library preparations were sequenced on an Illumina HiSeq 2000 platform by Novogene Bioinformatics Technology Co., Ltd. (Beijing, China), and 150-bp paired-end reads were generated.

A total of 3 μg of RNA per sample was used as input material for the sample preparation. Twelve sequencing libraries (i.e., three replicates per host species) were generated using the NEBNext${}^{{}^{®}}$ Ultra${}^{\mathrm{TM}}$ RNA Library Prep Kit for Illumina (NEB, Beijing, China) following the provided protocol. Clustering of the index-coded samples was then performed with the cBot Cluster Generation System using a TruSeq PE Cluster Kit v3-cBot-HS (Illumina, Beijing, China) according to the manufacturer’s instructions. After cluster generation, the library preparations were sequenced on an Illumina HiSeq 2000 platform by Novogene Bioinformatics Technology Co., Ltd. (Beijing, China), and 150-bp paired-end reads were generated.

### 2.3. Preprocessing of Illumina Reads and De Novo Transcriptome Assembly

Raw reads were filtered using Trimmomatic [25] to obtain high-quality, clean reads for assembly: adapter sequences, reads containing poly-N runs (≥10%), and low quality (sQ ≤ 5) reads were removed. The number of sequence duplications, GC content, Q20, and Q30 was then calculated for the clean reads. All downstream analyses were based on clean data. The clean data of all samples were uploaded to the Sequence Read Archive (SRA) with accession number PRJNA851568. De novo assembly of the transcriptome from the RNA-seq data was performed using the Trinity software package [26]), with min_kmer_cov set to two and default values for other parameters. Benchmarking Universal Single-Copy Orthologues (BUSCO) provides measures for quantitative assessment of transcriptome completeness based on evolutionarily informed expectations of gene content from conserved single-copy orthologs [27]. In this study, we evaluated the completeness of the assembly using BUSCO based on the embryophyta_odb10 database.

### 2.4. Functional Annotation of the Transcriptome

To annotate the assembled *T. nigrans* transcriptome, transcripts were aligned against the eggNOG Nr, and Swiss-Prot databases using BLASTP with a significance threshold of E ≤ 10${}^{-5}$. For functional categorization, the Gene Ontology (GO) and Kyoto Encyclopedia of Genes and Genomes (KEGG) pathways were analyzed and annotated using the TBtools [28].

### 2.5. Differential Expression Analysis and Co-Expression Network Analysis

To estimate the abundance of the de novo assembled transcripts, RSEM [29] was used; this protocol assesses transcript abundance based on the mapping of RNA-seq reads to the assembled transcriptome. The DESeq2 and edgeR in IDEAMEX [30] were used to identify differentially expressed genes (DEGs) among the host species. To minimize the false positive rate, only transcripts with threshold *p* values < 0.05 and absolute values of $\log_{2}\left(fold\;change\right)$ > 1 (as screened by IDEAMEX) were considered differentially expressed. The differentially expressed transcripts were divided into six sets, each comparing two host species, as follows: set I (Bpap vs. Cfor), set II (Bpap vs. Csep), set III (Bpap vs. Gbil), set IV (Cfor vs. Csep), set V (Cfor vs. Gbil), and set VI (Csep vs. Gbil). Pearson correlation coefficients were then calculated for all sets using gene expression data. A GO term enrichment analysis and KEGG pathways were performed for the DEGs using TBtools. We performed a co-expression network analysis using the WGCNA package [31] based on $\log_{2}\left(FPKM+1\right)$ values.

## 3. Results

### 3.1. Transcriptome Sequencing and De Novo Assembly of *T. nigrans* Transcriptome

Fresh leaves of *T. nigrans* collected from four host species were used to construct cDNA libraries, which were subsequently sequenced on an Illumina HiSeq 4000 platform. For each host species, three biological replicates were analyzed. After filtering the raw sequencing data, a total of 5,553,239,700 (Bpap-001), 6,299,403,300 (Bpap-002), 4,693,123,881 (Bpap-003), 4,183,362,900 (Cfor-001), 5,784,479,700 (Cfor-002), 5,156,214,900 (Cfor-003), 4,763,816,700 (Csep-001), 4,571,633,850 (Csep-002), 5,665,991,700 (Csep-003), 5,116,978,800 (Gbil-001), 4,066,401,900 (Gbil-002), and 4,932,610,200 (Gbil-003) clean bases were obtained (Table 1). The de novo assembly yielded 332,439 transcripts, with an N50 size of 1790 bp (Table 2). The BUSCO analysis showed 92.3% of 1614 single-copy genes in the embryophyta_odb10 database (Table 2). The results of the functional annotations of all transcripts are shown in Appendix A.

### 3.2. Analysis of Differentially Expressed Genes (DEGs) among Different Hosts

To identify DEGs among the host species, six pairwise comparisons were performed (Bpap vs. Cfor, Bpap vs. Csep, Bpap vs. Gbil, Cfor vs. Csep, Cfor vs. Gbil, and Csep vs. Gbil) using DESeq2 and edgeR in IDEAMEX. After combining the two methods (DESeq2 and edgeR), a total of 1526, 1093, 617, 352, 133, and 57 DEGs were detected for each of these comparisons, respectively (Figure 2A). Among the DEGs, a total of 1253, 864, 517, 259, 95, and 40 DEGs were functionally annotated, respectively, for each comparison (Appendix A). The principal components analysis (PCA) analysis of each comparison was shown in Figure 3 based on the IDEAMEX results.

After identifying the DEGs for each pairwise comparison, the subset of DEGs common to all samples was then determined. A total of 415 common unigenes were identified based on DESeq2 and degeR; these are presented in a Venn diagram and volcano map (Figure 2B,D). Partially common unigenes encoding putative transcription factors, including members of the cytochrome P450 family, HSP, ABC transporter family, UDP-glycosyltransferase family, and zinc finger families (Appendix A). In addition, unique DEGs were identified for the Bpap vs. Cfor comparison (*n* = 808), Bpap vs. Csep comparison (*n* = 329), Bpap vs. Gbil comparison (*n* = 87), Cfor vs. Csep comparison (*n* = 108), Cfor vs. Gbil comparison (*n* = 32), and Csep vs. Gbil comparison (*n* = 23), respectively (Figure 2A).

### 3.3. GO and KEGG Enrichment Analysis

To further explore differential gene expression in *T. nigrans*, the DEGs (from all comparisons) were used for GO and KEGG enrichment analyses. For the shared DEGs (415 across all comparisons), the GO enrichment analysis showed that a majority of DEGs belonged to the following categories: “UDP-glucosyltransferase activity” (GO:0035251), “transcription regulator activity” (GO:0140110), “ubiquitin-protein transferase activity” (GO:0004842), “chloroplast” (GO:0009507), “plant-type vacuole” (GO:0000325), “plasmodesma” (GO:0009506), “response to abiotic stimulus” (GO:0009628), “response to abscisic acid” (GO:0009737), “rhythmic process” (GO:0048511), “terpenoid metabolic process” (GO:0006721) (Figure 2C and Appendix A). The top five pathways identified in the KEGG enrichment analysis were: Circadian rhythm, cytochrome P450, organismal systems, environmental adaptation, and metabolism of terpenoids and polyketides (Table 3 and Appendix A). The unique DEGs identified in the six comparisons were subjected to GO and KEGG enrichment analyses. The most highly represented GO terms in Bpap vs. Cfor comparisons were: “phenylpropanoid metabolic process” (GO:0009698), “flavonoid metabolic process” (GO:0009812), “response to acid chemical” (GO:0001101), "fruit development" (GO:0010154), “seed development” (GO:0048316), “terpenoid biosynthetic process” (GO:0016114), “response to alcohol” (GO:0097305), “UDP-glycosyltransferase activity” (GO:0008194), “plant-type cell wall” (GO:0009505), and “long-chain fatty acid metabolic process” (GO:0001676) (Appendix A). In the KEGG analysis, the top five pathways were: Flavonoid biosynthesis, stilbenoid/diarylheptanoid/gingerol biosynthesis, phenylpropanoid biosynthesis, metabolism of terpenoids and polyketides, and amino sugar and nucleotide sugar metabolism (Appendix A). For specific DEGs in the Bpap vs. Csep comparison, significantly enriched GO terms included “transmembrane signaling receptor activity” (GO:0004888), “inorganic anion transmembrane transporter activity” (GO:0015103), “signaling receptor activity” (GO:0038023), “anion transmembrane transporter activity” (GO:0008509), “transmembrane transporter activity” (GO:0022857), “NADH metabolic process” (GO:0006734), “cellular response to jasmonic acid stimulus” (GO:0071395), “ATP metabolic process” (GO:0046034), “response to fatty acid” (GO:0070542), “ribonucleoside triphosphate metabolic process” (GO:0009199), etc. (Appendix A). In the KEGG analysis, the top five enriched pathways were glycolysis/gluconeogenesis, brassinosteroid biosynthesis, plant–pathogen interaction, plant hormone signal transduction, and environmental information processing (Appendix A). The most highly represented GO terms in Bpap vs. Gbil comparison were “plasma membrane” (GO:0005886), “cellular response to acid chemical” (GO:0071229), “hormone-mediated signaling pathway” (GO:0009755), “response to acid chemical” (GO:0001101), “response to salt stress” (GO:0009651), “cellular response to hormone stimulus” (GO:0032870), “cellular response to lipid” (GO:0071396), “response to osmotic stress” (GO:0006970), “signal transduction” (GO:0007165), and “cellular response to endogenous stimulus” (GO:0071495) (Appendix A). For specific DEGs in Csep vs. Cfor comparison, significantly enriched GO terms included “transcription regulator activity” (GO:0140110), “DNA-binding transcription factor activity” (GO:0003700), “chloroplast” (GO:0009507), “cell junction” (GO:0030054), “cell wall” (GO:0005618), “defense response” (GO:0006952), “regulation of RNA metabolic process” (GO:0051252), “regulation of RNA biosynthetic process” (GO:2001141), “regulation of nucleic acid-templated transcription” (GO:1903506), “response to hormone” (GO:0009725) (Appendix A). The top five enriched KEGG pathways for unique Csep vs. Cfor comparison were cutin/suberine and wax biosynthesis, diterpenoid biosynthesis, metabolism of terpenoids and polyketides, plant hormone signal transduction, and carbon fixation in photosynthetic organisms (Appendix A). There were no GO and KEGG enrichments found in the Cfor vs. Gbil and Csep vs. Gbil comparisons.

### 3.4. Identification of DEGs Co-Expression Modules

We identified co-expressed gene sets based on DEGs that appeared at least once in six comparisons through a weighted gene co-expression network analysis (WGCNA). The WGCNA analysis revealed several major co-expression modules, namely the brown module, blue module, turquoise module, and grey module, respectively (Figure 4A). There was no significant correlation between modules and hosts, which may be a small amount of samples. Further, we associated each of the co-expression modules with four samples. The brown module was correlated with Bpap. The blue module was correlated with Cfor and Gbil, and the turquoise module was correlated with Gbil and Csep (Figure 4B). Subsequently, to further explore genes in related modules, the GO enrichment analyses were followed. The blue module associated with Cfor and Gbil showed enrichment of GO terms related to “DNA-binding transcription factor activity” (GO:0003700), “hydrolase activity, hydrolyzing O-glycosyl compounds” (GO:0004553), “oxidoreductase activity” (GO:0016705), “monooxygenase activity” (GO:0004497), “plant-type cell wall” (GO:0009505), “plasmodesma” (GO:0009506), “apoplast” (GO:0048046), “plant-type cell wall organization or biogenesis” (GO:0071669), “xylan biosynthetic process” (GO:0045492), “glucuronoxylan biosynthetic process” (GO:0010417), and “phenylpropanoid metabolic process” (GO:0009698) (Appendix A). The Bpap-associated brown modules represented GO terms related to “RNA binding” (GO:0003723), “nucleic acid binding” (GO:0003676), “translation factor activity, RNA binding” (GO:0008135), “calmodulin binding” (GO:0005516), “ribosome” (GO:0005840), “cytosol” (GO:0005829), “cytosolic small ribosomal subunit” (GO:0022627), “translation” (GO:0006412), “peptide metabolic process” (GO:0006518), “organonitrogen compound biosynthetic process” (GO:1901566), and “cellular response to xenobiotic stimulus” (GO:0071466) (Appendix A). In a similar way, the turquoise module associated with Gbil and Csep exhibited representation of terms, such as “transmembrane transporter activity” (GO:0022857), “glucosyltransferase activity” (GO:0046527), “water transmembrane transporter activity” (GO:0005372), “UDP-glucosyltransferase activity” (GO:0035251), “plastid” (GO:0009536), “chloroplast” (GO:0009507), “plasmodesma” (GO:0009506), “response to abiotic stimulus” (GO:0009628), “response to acid chemical” (GO:0001101), “response to abscisic acid” (GO:0009737), and “response to radiation” (GO:0009314) (Appendix A).

## 4. Discussion

### 4.1. Transcriptome Assembly and Annotation

Advances in transcriptome sequencing technology and data mining platforms have led to rapid progress in comparative transcriptomics for non-model, non-crop plants, such as parasitic plants [32,33,34,35,36,37,38]. Here, transcriptomes were generated and compared for *T. nigrans* plants parasitizing four different host species. High-throughput sequencing generated more than 405 million clean reads from all samples, yielding approximately 248,751 unigenes after assembly. The assembled transcriptome comprised mostly short transcripts, as seen for other de novo assembled plant transcriptomes, perhaps as a result of the assembly algorithm used [39,40]. The contig N50 was 1790 bp, indicating that the Trinity assembly was of high quality. The assembly and annotation of the *T. nigrans* transcriptome provides a valuable resource for investigating the processes and pathways involved in *T. nigrans* host adaptation. For example, transcriptomes produced for the parasitic plants *Arceuthobium sichuanense* and *Cuscuta campestris* were used to identify DEGs among samples [24,40].

### 4.2. Gene Categories Associated with Plant Development and Host Selection

For some parasitic plants, such as *Arceuthobium sichuanense* and *Cuscuta campestris*, the establishment of vascular connections with the host plant allows the acquisition of nutrients and solutes; this nutrient sink can significantly reduce the biomass of the host [40,41]. In this study, based on the results of GO and KEGG enrichment, a large number of common unigenes (shared across *T. nigrans* individuals) were primarily related to carbohydrate metabolism, including glycolysis/gluconeogenesis; pyruvate metabolism; glyoxylate and dicarboxylate metabolism; and amino sugar and nucleotide sugar metabolism. Although the transcriptomic data were not directly related to the parasitic properties of *T. nigrans*, studies of other parasitic plants have shown that metabolism is related to parasitism [40,41,42,43]. In addition, unigenes involved in the energy metabolism in photosynthetic organisms (i.e., oxidative phosphorylation and carbon fixation) were also enriched in *T. nigrans*. We also found that part of the unigenes related to the circadian rhythm, cytochrome P450, metabolism of terpenoids and polyketides, and response to abscisic acid. Cytochrome P450 may contribute to environmental and host adaptation [44,45]. Abscisic acid is thought to be involved in the development [46,47]. The observed results may thus reflect the adaptive adjustment of *T. nigrans* to host selection and the environment. Furthermore, unigenes were identified specific to the Bpap vs. Cfor comparison (n = 808), Bpap vs. Csep comparison (n = 329), Bpap vs. Gbil comparison (n = 87), Cfor vs. Csep comparison (n = 108), Cfor vs. Gbil comparison (n = 32), and Csep vs. Gbil comparison (n = 23), most of which were annotated using the Gene Ontology and KEGG databases. In all annotated comparisons, most of the genes were related to the biosynthesis and metabolism of secondary metabolites. Unique genes of each comparison showed similarities in functional enrichment, such as Bpap vs. Cfor and Csep vs. Cfor comparisons involving the metabolism of terpenoids and polyketides; in Bpap vs. Csep and Csep vs. Cfor comparisons, we found genes related to plant hormone signal transduction that may be related to host differences. In addition, partial unigenes were associated with the plasma membrane, response to endogenous stimuli, ion binding, and organic hydroxy compound metabolic processes. Previous studies have shown that these factors are associated with plant defenses [48,49,50]. We identified co-expressed gene sets that were expressed in four samples via the weighted gene co-expression network analysis (WGCNA), a method widely used in the comparative transcriptome analysis [51,52,53,54]. These results indicate that each sample was associated with one or two co-expression modules that reflected the gene regulatory processes specific to each sample. We anticipate that the comprehensive information presented here will serve as a crucial resource to understand host differences and commonalities.

## 5. Conclusions

In summary, RNA-seq analysis revealed both host-specific and common pathways involved in *T. nigrans* parasitism; these results provide a foundation for further study of the molecular factors and functions underlying adaptation to different host plants. The putative key genes and pathways identified here may also be used to explore the molecular factors and mechanisms explaining host diversity. The analysis represents a critical step in promoting the development of molecular ecology tools for parasite–host systems and sustainable integrated management programs.

## Figures and Tables

**Figure 1 genes-13-01173-f001:**
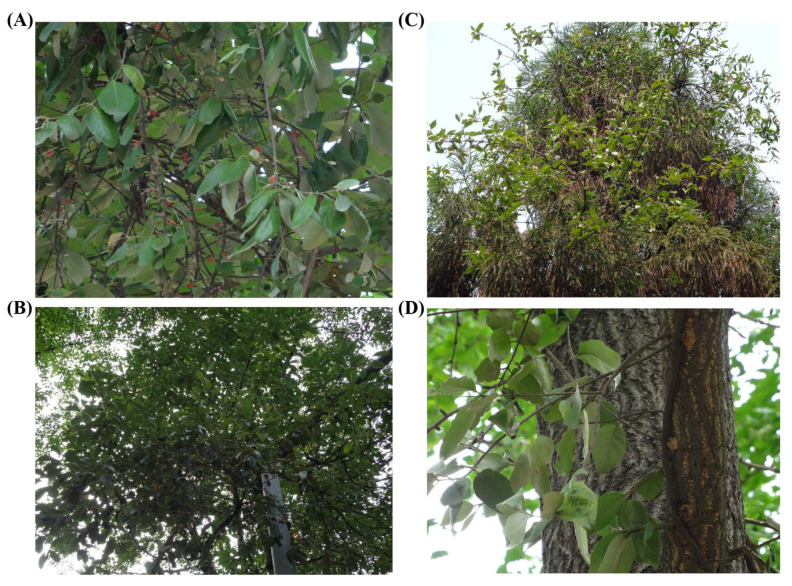
The host plants were ’parasited’ by *T. nigrans*. (**A**) Broussonetia papyrifera. (**B**) Cryptomeria fortunei. (**C**) Cinnamomum septentrionale. (**D**) Ginkgo biloba.

**Figure 2 genes-13-01173-f002:**
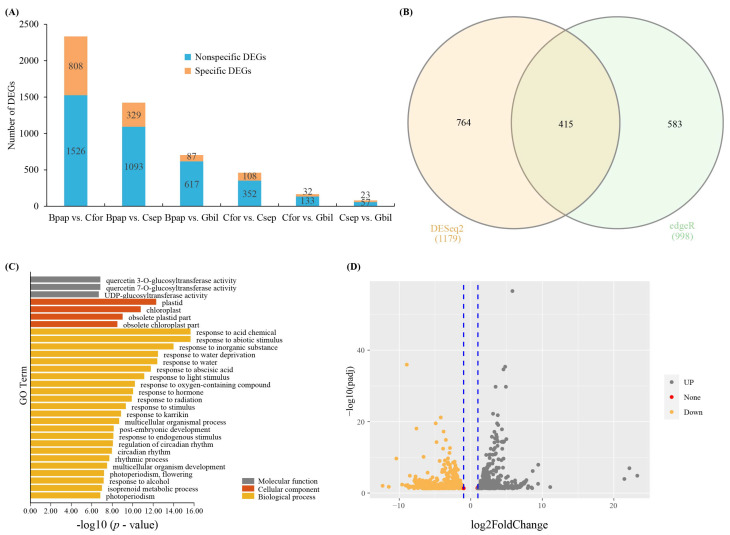
Summary of differentially expressed genes (DEGs). (**A**) The number of DEGs for each pairwise comparison. (**B**) Common unigenes based DESeq2 and edgeR. (**C**) GO enrichment of common unigenes. (**D**) The volcano map DEGs.

**Figure 3 genes-13-01173-f003:**
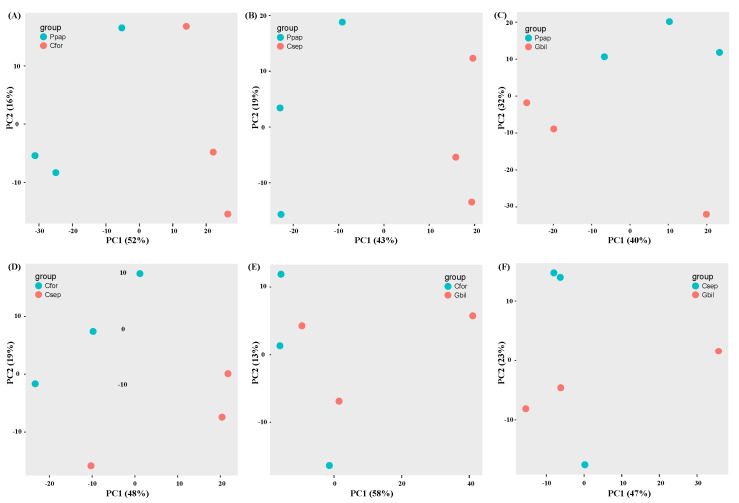
PCA analysis of each comparison. (**A**) Represented Bpap vs. Cfor comparison. (**B**) Represented Bpap vs. Csep comparison. (**C**) Represented Bpap vs. Gbil comparison. (**D**) Represented Cfor vs. Csep comparison. (**E**) Represented Cfor vs. Gbil comparison. (**F**) Represented Csep vs. Gbil comparison.

**Figure 4 genes-13-01173-f004:**
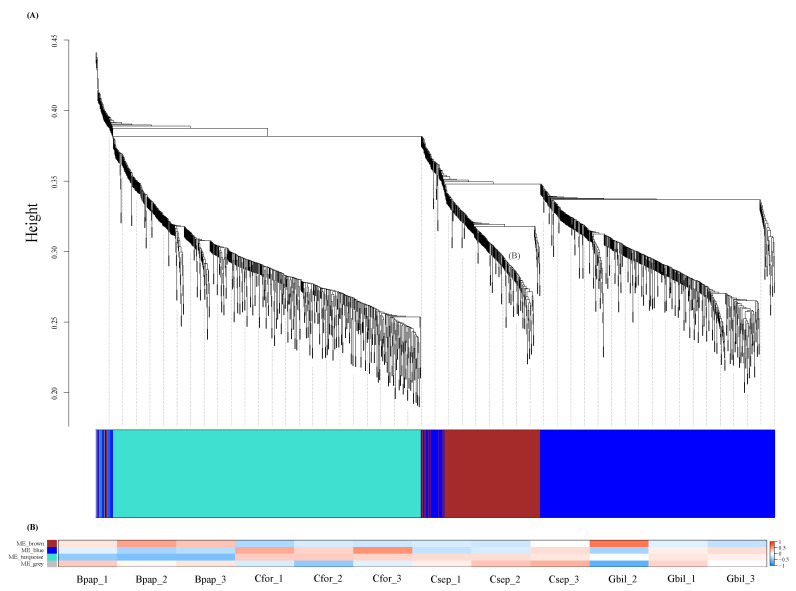
Weighted gene co-expression network analysis (WGCNA) in all samples. (**A**) Hierarchical clustering tree (dendrogram) of unigenes. (**B**) Heatmap showing the comparison of modules among all samples.

**Table 1 genes-13-01173-t001:** Summary of *T. nigrans* RNA-sequencing data analyzed in this study.

Sample ID	Raw Reads	Clean Reads	Bases (bp)
Bpap-001	37,814,806	37,021,598	5,553,239,700
Bpap-002	42,783,200	41,996,022	6,299,403,300
Bpap-003	34,569,496	31,134,696	4,693,123,881
Cfor-001	29,838,218	27,889,086	4,183,362,900
Cfor-002	39,435,952	38,563,198	5,784,479,700
Cfor-003	34,963,898	34,374,766	5,156,214,900
Csep-001	32,646,192	31,758,778	4,763,816,700
Csep-002	32,698,236	30,735,719	4,571,633,850
Csep-003	38,673,528	37,773,278	5,665,991,700
Gbil-001	34,908,518	34,113,192	5,116,978,800
Gbil-002	27,899,308	27,109,346	4,066,401,900
Gbil-003	33,457,780	32,884,068	4,932,610,200
Total	419,689,132	405,353,747	40,058,127,750

**Table 2 genes-13-01173-t002:** Overview of the de novo transcriptome assembly.

	Transcripts	Unigenes
Total number	332,439	234,510
Median contig length (bp)	431	348
Minimum length (bp)	182	201
Maximum length (bp)	27,616	27,616
N50 (bp)	1790	729
Number, ≤500 bp	187,734	165,956
Number, >500 bp	144,705	68,554
Total nucleotides	296,766,287	136,260,562
BUSCO (%)	92.3	65.5

**Table 3 genes-13-01173-t003:** KEGG enrichment analysis of common DEGs.

Class	KEGG Description	*p* Value
Organismal systems	Circadian rhythm—plant	6.34 × 10${}^{-9}$
Brite hierarchies	Cytochrome P450	1.14 × 10${}^{-5}$
Organismal systems	Organismal Systems	2.35 × 10${}^{-5}$
Organismal Systems	Environmental adaptation	2.35 × 10${}^{-5}$
Brite hierarchies	Transcription factors	9.82 × 10 ${}^{-5}$
Metabolism	Carotenoid biosynthesis	2.18 × 10${}^{-4}$
Metabolism	Metabolism of terpenoids and polyketides	2.41 × 10${}^{-4}$
BRITE hierarchies	Transporters	0.003102542
Not included in the pathway or BRITE	Signaling proteins	0.003192151
BRITE hierarchies	Protein families: metabolism	0.004501004
BRITE hierarchies	Chaperones and folding catalysts	0.004860346
Environmental information processing	Plant hormone signal transduction	0.006261807
BRITE hierarchies	Protein phosphatases and associated proteins	0.019453528
Environmental information processing	Environmental information processing	0.021745631
Environmental information processing	Signal transduction	0.021941008
Not included in pathway or BRITE	Unclassified: signaling and cellular processes	0.032328609
BRITE hierarchies	Ubiquitin system	0.039933933
Environmental information processing	6 MAPK signaling pathway—plant	0.042543616

## Data Availability

The data presented in this study are available upon request from the corresponding author.

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
