# Peer review of "Transcriptome Profiling of a Common Mistletoe Species Parasitizing Four Typical Host Species in Urban Southwest China"

_genes, 2022, doi:10.3390/genes13071173_

Round 1
Reviewer 1 Report
In this paper, Kuang and colleagues report a comparative transcriptome analysis in Taxillus nigrans, comparing transcriptional profiles among four hosts.
It is a typical RNA-seq analysis. My major concerns in this manuscript are:
- The lack of a deposit of raw data in a repository and I also strongley recommend to make unigene accessible through a public link.
- Since authors do not report qPCR validation of differentially expressed genes, authors must run several more bioinformatic analyses to guarantee the quality of their findings, including:
- A BUSCO analysis to check transcriptome completeness (https://busco.ezlab.org/)
- The use of consensus differentially expressed genes, using more than one tool. Authors might use IDEAMEX (https://doi.org/10.3389/fgene.2019.00279) or consensusDE (https://peerj.com/articles/8206/)
- The presentation of GO characterization on the main paper
- PCA analysis of libraries
- A table of the 10 or 20 most important genes that were differentially expressed in all situations
- A coexpression analysis to detect functional hubs among hosts. Authors might use WGCNA (https://horvath.genetics.ucla.edu/html/CoexpressionNetwork/Rpackages/WGCNA/), CemiTool (https://www.bioconductor.org/packages/release/bioc/html/CEMiTool.html), or Clust (https://github.com/BaselAbujamous/clust)
After these inclusions (or q PCR validation), manuscript can be reassessed.
Minor issues:
- line 9: instead of 40.06Gb report the >400M reads
- line 9: nine or twelve cDNA libraries?
- lines 38-48: this part can be reduced
- lines 50-57: this part should be extended
- section 2.1: can authors provide pictures of the plant in its distinct hosts?
- section 2.3: how RPKM was calculated? please, state programs and parameters. Moreover, consider using TPM instead of RPKM.
After these changes, the manuscript can be re-evaluated.
Reviewer 2 Report
The manuscript entitled "Transcriptome profiling of a common mistletoe species parasitizing four typical host species in urban southwest China" makes an excellent contribution toward understanding the molecular mechanisms in host × parasite interaction. The authors are requested to revise the manuscript by paying attention to the following comments:
1. The introduction is very limited. While its provided sufficient background on parasitic plants, it did not however highlight the economic importance of the studied species. In addition, the authors need to strengthen the introduction with more global literature on transcriptome analysis and how relevant are these analysis in the current context.
2. The stage of the T. nigrans parasitism was not specified in the M&M. Authors are kindly requested to specify this as it will help draw conclusions on the data (severity of the parasite differs depending on parasite developmental stages: as a hemi-parasite, the parasite is expected to siphon more in early stage than later!).
In addition, transcriptome analysis on different tissue would lead to different patterns or would have been good to include the point de attachment (haustorium or root segment of the parasite). This would have allow to understand responses in the most part of the parasitism as well as the variation in response to specific host!
3. Line 43: Please keep "Thesium chinense" in Italic!
4. Line 148-149: The authors stated as follow: "A total of 381 common unigenes were identified in all six pairwise comparisons; these are presented in a Venn diagram". I might be wrong but I could not see any Figure related to this, neither any Venn diagram. Please provide Figure # (main or in suppl.)
5. Line 214-215: It seems quite strange that only 2 genes were found to be differentially (DE) in Csep vs. Gbil parasitism. In addition, the authors did not provide any further potential explanation for this. What could be the main biological drivers for such difference ? Or is that simply related to technical/analytical errors.
6. Results need to be condensed/well formulated and Discussion needs to be strengthened with more details and supporting explanations. In Results/Discussion sections, is there a more reader-friendly way to present this information? The excessive numbers of reads, GO make the flow below optimum. Perhaps the authors need a better Table distribution so improve text flow.
Round 2
Reviewer 1 Report
Authors addressed most issues.
However, some issues are still needed to be addressed:
line 9 - Mb should be replaced to million
BUSCO analysis should be clearly explained in methods.
In Response 1: Thank you for your valuable suggestions. We have uploaded the unigenes to NGDC and are currently reviewing. Therefore, the link of the data will be released as soon as possible.
Reply: It is mandatory to include RAW data in SRA-NCBI and/or ENA-EMBL databases, besides NGDC to unigenes (which accession is also mandatory to be present in the manuscript)
